# The Quenching-Activation Behavior of the Gradient Descent Dynamics for Two-layer Neural Network Models

## Abstract

A numerical and phenomenological study of the gradient descent (GD) algorithm for training two-layer neural network models is carried out for different parameter regimes. It is found that there are two distinctive phases in the GD dynamics in the under-parameterized regime: An early phase in which the GD dynamics follows closely that of the corresponding random feature model, followed by a late phase in which the neurons are divided into two groups: a group of a few (maybe none) "activated" neurons that dominate the dynamics and a group of "quenched" neurons that support the continued activation and deactivation process. In particular, when the target function can be accurately approximated by a relatively small number of neurons, this quenching-activation process biases GD to picking sparse solutions. This neural network-like behavior is continued into the mildly over-parameterized regime, in which it undergoes a transition to a random feature-like behavior where the inner-layer parameters are effectively frozen during the training process. The quenching process seems to provide a clear mechanism for "implicit regularization". This is qualitatively different from the GD dynamics associated with the "mean-field" scaling where all neurons participate equally.

## 1 Introduction

In the past few years, much effort has been devoted to the understanding of the theoretical foundation behind the spectacular success of neural network (NN)-based machine learning. The main theoretical questions concern the training process and the generalization property of solutions found. For two-layer neural network (2LNN) and deep residual neural network models, it has been proved that solutions with "good" generalization performance do exist. Specifically, it has been shown that for the appropriate classes of target functions, the generalization error associated with the global minimizers of some *properly regularized* 2LNN and deep residual neural networks obey Monte Carlo-like estimates: $O(1/m)$ for the approximation error and $O(1/\sqrt{n})$ for the estimation error, where $m$ and $n$ are the number of parameters and the size of the training set, respectively (Barron, 1994; Bach, 2017; E et al., 2019b;a). The fact that these estimates do not suffer from the curse of dimensionality (CoD) is one of the fundamental reasons behind the success of neural network models in high dimensions.

An important open question is: Do standard optimization algorithms used in practice find good solutions? NN-based models often work in the over-parameterized regime where the models can easily fit all the training data, and some of solutions may give rise to large test errors (Wu et al., 2017). However, it has been observed in practice that small test error can often be achieved with appropriate choice of the hyper-parameters, even without the need of explicit regularization (Neyshabur et al., 2014; Zhang et al., 2017). This means that there are some "implicit regularization" mechanisms at work for the optimization algorithm with the particular choices of hyper-parameters.

A rather complete picture has been established for highly over-parameterized NN models. Unfortunately the overall result is somewhat disappointing: While one can prove that GD converges to a global minimizer of the empirical risk (Du et al., 2019b;a), the generalization properties of this global minimizer is no better than that of an associated random feature model (RFM) (Jacot et al., 2018; E et al., 2020; Arora et al., 2019). In fact, E et al. (2020) and Arora et al. (2019) proved that the entire GD paths for the NN model and the associated RFM stay uniformly close for all time.

A natural question is then: Can there be implicit regularization when the network is less over-parameterized? What would be the mechanism of the implicit regularization? More generally, what is the qualitative behavior of the GD dynamics in different regimes including the under-parameterized regimes? In this paper, we provide a systematic investigation for two-layer neural networks by well-designed experiments. Our objective is to get some insight from this kind of experimental studies, which we hope will be helpful for subsequent theoretical work. Specifically, our findings are summarized as follows.

- It is observed that when the network is less over-parameterized, the GD dynamics exhibit two phases. During the first phase, GD follows closely that of the corresponding RFM. Afterwards, GD enters the second phase where neurons form two groups (the first group might be empty): a group of activated neurons and a group of quenched neurons. Depending on the target functions, the quenched neurons can exhibit continued quenching and sparse activation processes. In particular, if the target function can be well approximated by a small number of neurons, GD is biased to picking sparse solutions.

- Based on these observations, we then investigate how the extent of over-parameterization affects the generalization properties of GD solutions. We find that the test error shows a sharp transition within the *mildly over-parameterized* regime. This transition suggests that implicit regularization is quite sensitive to the change of the network width.

- Lastly, we study 2LNNs under mean-field scaling (Chizat & Bach, 2018; Mei et al., 2018; Rotskoff & Vanden-Eijnden, 2018; Sirignano & Spiliopoulos, 2020), i.e. with an extra $1/m$ factor added to the expression of the function, where $m$ denotes the number of neurons. We observe that in this case all the neurons contribute pretty much equally and the test performance is much more robust to the change of network width.

## 2 PRELIMINARIES

### 2.1 TWO-LAYER NEURAL NETWORKS

Under *conventional* scaling, a two-layer neural network model is given by:

$$f_m(\boldsymbol{x}; \boldsymbol{a}, \boldsymbol{B}) = \sum_{j=1}^{m} a_j \sigma(\boldsymbol{b}_j^T \boldsymbol{x}) = \boldsymbol{a}^T \sigma(\boldsymbol{B}\boldsymbol{x}), \tag{1}$$

where $\boldsymbol{a} \in \mathbb{R}^m, \boldsymbol{B} = (\boldsymbol{b}_1, \boldsymbol{b}_2, \ldots, \boldsymbol{b}_m)^T \in \mathbb{R}^{m \times d}$ and $\sigma(t) = \max(0, t)$ is the ReLU activation function. Later we will consider the *mean-field* scaling where the expression above is replaced by

$$f_m(\boldsymbol{x}; \boldsymbol{a}, \boldsymbol{B}) = \frac{1}{m} \sum_{j=1}^{m} a_j \sigma(\boldsymbol{b}_j^T \boldsymbol{x}) = \frac{1}{m} \boldsymbol{a}^T \sigma(\boldsymbol{B}\boldsymbol{x}), \tag{2}$$

but we will focus on the conventional scaling unless indicated otherwise. As a comparison, the random feature model is given by $f_m(\boldsymbol{x}; \boldsymbol{a}, \boldsymbol{B}_0)$, where only the coefficient $\boldsymbol{a}$ can be varied; $\boldsymbol{B}_0$ is randomly sampled and is fixed during training.

Let $S = \{(\boldsymbol{x}_i, y_i = f^*(\boldsymbol{x}_i))\}_{i=1}^{n}$ denote the training set. $f^*$ is the target function. We assume that $\{\boldsymbol{x}_i\}$ are drawn independently from $\pi_0$, the uniform distribution over $\mathbb{S}^{d-1} := \{\boldsymbol{x} \in \mathbb{R}^d : \|\boldsymbol{x}\| = 1\}$. The empirical risk and the population risk are defined by $\hat{\mathcal{R}}_n(\boldsymbol{a}, \boldsymbol{B}) = \frac{1}{n} \sum_{i=1}^{n} (f_m(\boldsymbol{x}_i; \boldsymbol{a}, \boldsymbol{B}) - f^*(\boldsymbol{x}_i))^2$ and $\mathcal{R}(\boldsymbol{a}, \boldsymbol{B}) = \mathbb{E}_{\boldsymbol{x} \sim \pi_0}[(f_m(\boldsymbol{x}; \boldsymbol{a}, \boldsymbol{B}) - f^*(\boldsymbol{x}))^2]$, respectively.

Following the study of the function space for two-layer neural networks (E et al., 2019c; Bach, 2017), we will focus on target functions of the form $f^*(\boldsymbol{x}) = \mathbb{E}_{\boldsymbol{b} \sim \pi^*}[a^*(\boldsymbol{b})\sigma(\boldsymbol{b}^T \boldsymbol{x})]$ with $\pi^*$ being a probability distribution over $\mathbb{S}^{d-1}$. The population risk can be written as:

$$\mathcal{R}(\boldsymbol{a}, \boldsymbol{B}) = \mathbb{E}_{\boldsymbol{x}}[(\sum_{j=1}^{m} a_j \sigma(\boldsymbol{b}_j \cdot \boldsymbol{x}) - \mathbb{E}_{\boldsymbol{b} \sim \pi^*}[a^*(\boldsymbol{b})\sigma(\boldsymbol{b}^* \cdot \boldsymbol{x})])^2]$$

$$= \sum_{j_1, j_2=1}^{m} a_{j_1} a_{j_2} k(\boldsymbol{b}_{j_1}, \boldsymbol{b}_{j_2}) - 2 \sum_{j=1}^{m} a_j \mathbb{E}_{\boldsymbol{b} \sim \pi^*}[a^*(\boldsymbol{b})k(\boldsymbol{b}_j, \boldsymbol{b})]$$

$$+ \mathbb{E}_{\boldsymbol{b} \sim \pi^*} \mathbb{E}_{\boldsymbol{b}' \sim \pi^*}[a^*(\boldsymbol{b})a^*(\boldsymbol{b}')k(\boldsymbol{b}, \boldsymbol{b}')], \tag{3}$$

where $k(\boldsymbol{b}, \boldsymbol{b}') = \|\boldsymbol{b}\|\|\boldsymbol{b}'\| \left(\sin\theta + (\pi - \theta)\cos\theta\right)$ with $\theta = \arccos(\langle \hat{\boldsymbol{b}}, \hat{\boldsymbol{b}}' \rangle)$ (Cho & Saul, 2009).

Since our primary interest is to gain some insight about the behavior of the training dynamics in different parameter and scaling regimes, we choose to work with an idealized setting with simple synthetic target functions instead of real data sets. Real data sets have extra complicating factors that prevent us from focusing on particular aspects of the training process. In fact most of our efforts will be devoted to target functions of finite neurons or that can be accurately approximated by a relatively small number of neurons, similar to the widely used teacher-student setting (Saad & Solla, 1995; Tian, 2017; Safran & Shamir, 2018; Goldt et al., 2019). However, we will also discuss one example for which this is not the case.

## 2.2 GD DYNAMICS FOR TWO-LAYER NEURAL NETWORKS

We consider the most popular initialization (LeCun et al., 2012; He et al., 2015),

$$a_j(0) \sim \mathcal{N}(0, \beta^2), \qquad \boldsymbol{b}_j(0) \sim \mathcal{N}(0, I/d). \tag{4}$$

For the original LeCun initialization (LeCun et al., 2012), $\beta = 1/\sqrt{m}$. However, we have found consistently that for 2LNNs the behavior of the GD dynamics is qualitatively very close to the case when $\beta = 0$. We refer to Appendix A for some numerical results along this line. For simplicity we will focus on the case when $\beta = 0$.

Du et al. (2019b) proved that GD converges exponentially fast to a global minimizer in the highly over-parameterized regime. Subsequently it was shown in (E et al., 2020) that these GD solutions are uniformly close to that of the associated RFM with $\boldsymbol{B}_0 = \boldsymbol{B}(0)$ as the features:

**Theorem 1** (Informal). *Let $\tilde{\boldsymbol{a}}(t)$ denote the GD solution of associated RFM. For any $\delta \in (0, 1)$, if $m \geq \text{poly}(n, \log(1/\delta)), \beta = 0$, with probability $1 - \delta$ we have*

$$\sup_{\boldsymbol{x} \in \mathbb{S}^{d-1}, t \in [0, \infty)} |f_m(\boldsymbol{x}; \boldsymbol{a}(t), \boldsymbol{B}(t)) - f_m(\boldsymbol{x}; \tilde{\boldsymbol{a}}(t), \boldsymbol{B}_0)| \leq \frac{\text{poly}(n, \log(1/\delta))}{\sqrt{m}}. \tag{5}$$

The key observations for this case is the time scale separation: $\dot{a}_j(t) \sim O(\|\boldsymbol{b}_j\|) = O(1)$ and $\dot{\boldsymbol{b}}_j(t) \sim O(|a_j|) = O(\text{poly}(n)/m)$. In the highly over-parameterized regime, the dynamics of $\boldsymbol{b}_j$ is effectively frozen. GD for the 2LNN degenerates to the GD for the corresponding RFM.

There are three important large parameters: $m, n, d$. They are the number of neurons, the number of training samples, the input dimension, respectively. There are two obvious extreme situations that are of interest. One is when $m \gg n$. This was described above in Theorem 1 and is relatively well understood. The other is when $n \gg m$, which will be investigated by considering $n = \infty$. The regime when $m \sim n$ or $m \sim n/(d+1)$ are also of interest since these are regimes where the "resonance" (or the closely related "double descent") phenomena might occur, as we learned from previous work on RFM (Advani & Saxe, 2017; Belkin et al., 2019; Ma et al., 2020).

## 3 GD DYNAMICS FOR THE CASE WITH INFINITE SAMPLES

As a starting point, we first investigate the GD dynamics for the population risk, i.e. $n = \infty$. We will see later that the phenomena revealed here are indicative of the neural network-like (NN-like) behavior for GD dynamics.

### 3.1 SINGLE NEURON TARGET FUNCTION: NEURON ACTIVATION AND DEACTIVATION

First we look at the case where the target function is a single neuron:

$$f_1^*(\boldsymbol{x}) = \sigma(\boldsymbol{b}^* \cdot \boldsymbol{x}),$$

where $\boldsymbol{b}^* = \boldsymbol{e}_1$. The results are presented in Figure 1. One can observe several interesting features about the GD dynamics.

(1) Initially, the GD dynamics for the 2LNN is close to that of the corresponding RFM. (2) The GD dynamics for the 2LNN and RFM depart from each other around the time when the loss function for the RFM starts to saturate. (3)The converged solution for the 2LNN is very sparse. In fact only one neuron contributes significantly to the model in this experiment.

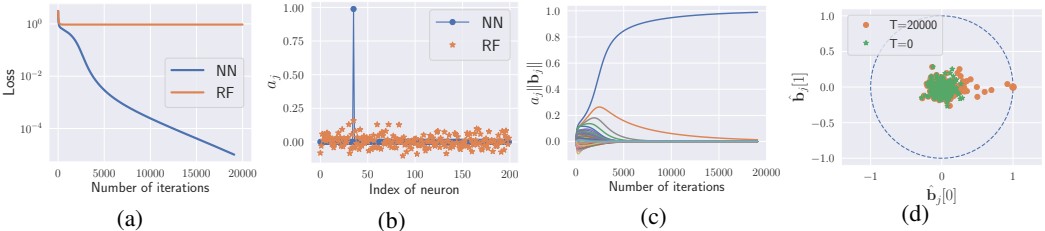

Figure 1: The dynamic behavior of learning the single neuron target function. The results of RFM are shown as a comparison. Here, $m = 200, d = 100$ and the learning rate is 0.001. (a) The dynamics of the population risk. (b) The $a$ coefficients of the converged solution. (c) The dynamics of the magnitude of each neuron, where the magnitude is defined as $a\|\boldsymbol{b}\|$. (d) The projection to the first two coordinates of $\hat{\boldsymbol{b}}$. The green and orange ones correspond to the initialization and GD solution, respectively.

We also see that there are two phases in the GD dynamics. In the first phase, the dynamics follows closely the GD dynamics for the RFM. In the second phase, the outer layer coefficients $a$ are small except for one neuron. In the transition from the first to the second phase, except for one neuron, all the other neurons are "quenched" in the sense that their outer layer coefficients $a$ keep decreasing. Thus, the dynamics of their inner layer weights $\boldsymbol{b}$ become very slow. Consequently, as shown in Figure 1d, only one neuron is "specialized" to the teacher $\boldsymbol{b}^*$ (Saad & Solla, 1995), and the inner layer weights of most neurons have hardly changed. The same behavior was observed with other realizations of the initial data, except that the number of "activated" neurons can be different. But in all cases, we always observe few "activated" neurons in the second phase.

### 3.1.1 CIRCLE NEURON TARGET FUNCTION: MULTI-STEP PHENOMENON

Next we consider a more sophisticated target function: $f_2^*(\boldsymbol{x}) = \mathbb{E}_{\boldsymbol{b}\sim\pi_2}[\sigma(\boldsymbol{b}^T\boldsymbol{x})]$, where $\pi_2$ is the uniform distribution over the unit circle $\Gamma = \{\boldsymbol{b} \in \mathbb{R}^d : b_1^2 + b_2^2 = 1 \text{ and } b_i = 0 \ \forall i > 2\}$.

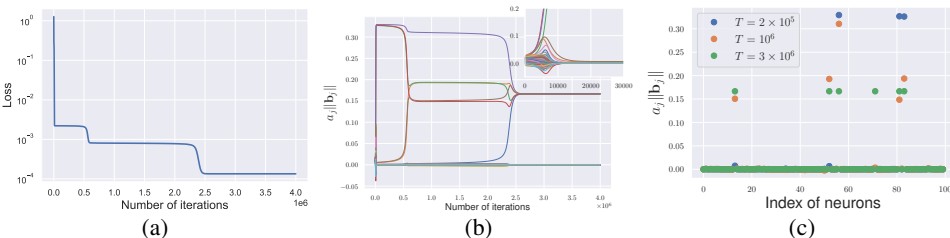

Figure 2: The dynamic behavior of learning the circle neuron target function. Here $m = 100, d = 100$ and the learning rate is 0.005. (a) The dynamics of the population risk. (b) The dynamics of the magnitude of each neuron; the inset is the zoom-in of the first 30000 iterations. (c) The $a$ coefficients of the solutions selected to represent the three steps.

A typical dynamic behavior of the population risk is shown in Figure 2a. We see that there are still two phases. But in the second phase, the population risk decreases in a "step-like" fashion. To see what happens, we plot the dynamics of the $a$ coefficient of each neuron in Figure 2b. We see that after the first phase, most of the neurons start to die out slowly (see the inset of Figure 2b). As the GD dynamics proceeds, a few new neurons are activated from the "quenched neurons". This activation process can be very slow, and the loss function is almost constant before activation actually happens. The activation process is relatively fast and causes a fast decay of the loss function. Figure 2c shows three representative solutions for the three steps. We see that from the first and second step, two more neurons are activated. From the second to the third step, one more neuron pops out.

### 3.1.2 FINITE NEURON TARGET FUNCTIONS

The observation that GD dynamics picks out sparse solutions and the associated quenching-activation behavior happens in a more general setting: learning finite neurons. Consider the target function $f_3^*(\boldsymbol{x}) = \sum_{j=1}^{m^*} a_j^* \sigma(\boldsymbol{b}_j^* \cdot \boldsymbol{x})$.

We are interested in the case: $m > m^*$. The single-neuron target function is a special case with $m^* = 1$. Figure 3 shows the dynamic behavior for $m = 50, m^* = 40$. In this case $m$ is the same order of $m^*$, but we still see the GD dynamics tends to find solutions with the number of active neurons close to $m^*$. The learning process is qualitatively similar to the case of circle neuron target function except that the activation process proceeds in a more continuous fashion and therefore the step-like behav-

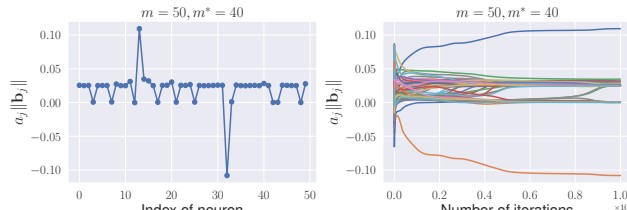

Figure 3: The dynamic behavior of learning a finite-neuron target function. Here, $a_j^* = 1/m$, $\{b_j^*\}$ are uniformly drawn from $\mathbb{S}^{d-1}$ with $d = 100$ and the learning rate is 0.001. **Left:** The magnitude of each neuron of the convergent solution. **Right:** The dynamics of the magnitude of each neuron.

ior is less pronounced. This trend becomes clearer if $m$ is much larger than $m^*$. See Appendix B for more experiments.

### 3.1.3 SURFACE NEURON TARGET FUNCTION

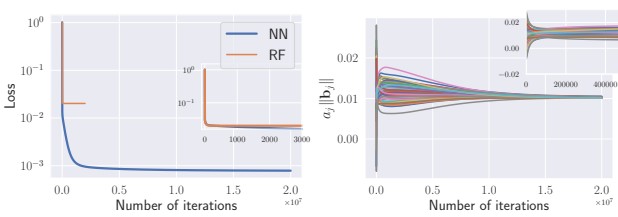

Figure 4: The dynamic behavior of learning the surface neuron target function. Here $m = 100, d = 100$ and learning rate $\eta = 0.005$. **Left:** The dynamics of the population risk; **Right:** The dynamics of the magnitude for each neuron. The inset is the zoom-in of the first $500,000$ iterations.

The target functions studied above are all functions that can be accurately approximated by a small set of neurons. Next we consider an example in the opposite direction: the "surface neuron target function"

$$f_4^*(\boldsymbol{x}) = \mathbb{E}_{\boldsymbol{b} \sim \pi_3}[\sigma(\boldsymbol{b}^T \boldsymbol{x})]$$

where $\pi_3$ is the uniform distribution over $\Omega = \{\boldsymbol{b} : \sum_{i=1}^{d/2} b_i^2 = 1 \text{ and } b_i = 0, \forall i > d/2\}$. This function is represented by a very large set of neurons, each of which contributes an equal small amount. Shown in Figure 4 are the numerical results.

One can see that in this case all the neurons are quenched and there are no activated neurons. The activation process is replaced by smooth changes of all the neurons.

### 3.2 UNDERSTANDING THE SECOND PHASES BY USING EFFECTIVE DYNAMICS

In the second phase, the neurons can be divided into two groups (the first group might be empty): $\mathbb{I}_1$ and $\mathbb{I}_2$, where $\mathbb{I}_1, \mathbb{I}_2$ are the set of indices of active and quenched neurons, respectively. We have $a_j = O(1)$ for $j \in \mathbb{I}_1$ and $a_j = o(1)$ for $j \in \mathbb{I}_2$. Denote by $f_{\mathbb{I}_i}(\boldsymbol{x}; t) = \sum_{j \in \mathbb{I}_i} a_j(t)\sigma(\boldsymbol{b}_j^T(t)\boldsymbol{x})$ with $i = 1, 2$ the functions represented by the active and quenched neurons, respectively.

The dynamics of neurons for the two groups are very different. For $j \in \mathbb{I}_1$, both $a_j$ and $\boldsymbol{b}_j$ changes significantly. However, for $j \in \mathbb{I}_2$, $a_j(t)$ evolves much faster than $\boldsymbol{b}_j(t)$, since $\dot{a}_j \sim O(\|\boldsymbol{b}_j\|) = O(1), \dot{\boldsymbol{b}}_j \sim O(|a_j|) \ll 1$. As a result $\{a_j\}_{j \in \mathbb{I}_2}$ is effectively slaved to the optimal solutions with $\{\boldsymbol{b}_j\}_{j \in \mathbb{I}_2}$ held fixed, which is given by

$$\operatorname{argmin}_{\{a_j\}_{j \in \mathbb{I}_2}} \mathbb{E}_{\boldsymbol{x}}[(\sum_{j \in \mathbb{I}_2} a_j\sigma(\boldsymbol{b}_j^T(t)\boldsymbol{x}) + f_{\mathbb{I}_1}(\boldsymbol{x}; t) - f^*(\boldsymbol{x}))^2]. \quad (6)$$

In this way, one obtains an effective dynamics governing the second phase: For active neurons, i.e. $j \in \mathbb{I}_1$,

$$\dot{a}_j(t) = -\mathbb{E}_{\boldsymbol{x}}[(f(\boldsymbol{x}; t) - f^*(\boldsymbol{x}))\sigma(\boldsymbol{b}_j^T(t)\boldsymbol{x})]$$
$$\dot{\boldsymbol{b}}_j(t) = -\mathbb{E}_{\boldsymbol{x}}[(f(\boldsymbol{x}; t) - f^*(\boldsymbol{x}))a_j(t)\sigma'(\boldsymbol{b}_j^T(t)\boldsymbol{x})\boldsymbol{x}]; \quad (7)$$

for quenched neurons, i.e. $j \in \mathbb{I}_2$,

$$a_j(t) = a_j^*(t)$$
$$\dot{\boldsymbol{b}}_j(t) = -\mathbb{E}_{\boldsymbol{x}}[(f(\boldsymbol{x}; t) - f^*(\boldsymbol{x}))a_j(t)\sigma'(\boldsymbol{b}_j^T(t)\boldsymbol{x})\boldsymbol{x}], \quad (8)$$

where $\{a_j^*(t)\}_{j \in \mathbb{I}_2}$ is the solution of (6), and $f(\boldsymbol{x}; t) = \sum_j a_j(t)\sigma(\boldsymbol{b}_j^T(t)\boldsymbol{x})$ is the functions represented by the solutions of the effective dynamics at the time $t$.

We numerically test how accurate the effective dynamics is for two target functions: single neuron and surface neuron. For surface neuron, $\mathbb{I}_1$ is empty. Shown in Figure 5 are the results. We see that indeed the effective dynamics is able to capture the original dynamics very well, even at relatively long-time scales (see Figure 13 in the appendix).

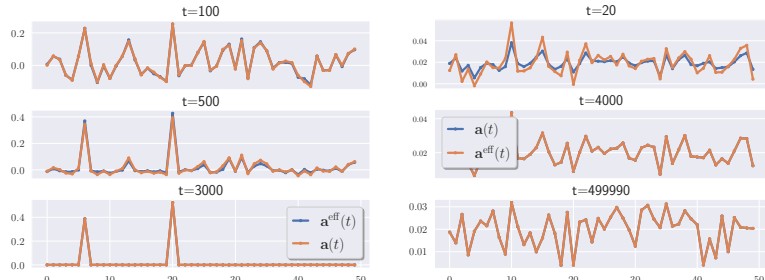

Figure 5: Comparison between $\boldsymbol{a}(t)$ and $\boldsymbol{a}^{\mathrm{eff}}(t)$ at different times, where $\boldsymbol{a}(t), \boldsymbol{a}^{\mathrm{eff}}(t)$ are the solutions of the original and effective dynamics, respectively. **Left:** The single neuron target function $f_1^*$. **Right:** The surface neuron target function $f_4^*$.

# 4 GD DYNAMICS FOR THE CASE OF FINITE TRAINING SAMPLES

We now turn to the more realistic situation where the size of the training set is finite.

## 4.1 THE DYNAMIC BEHAVIOR

We refer the reader to Appendix D for some numerical results of the highly over-parameterized regime as well as the under-parameterized regime, i.e. $m < n/(d+1)$. We find that in the under-parameterized regime, the dynamic behavior of GD is qualitatively similar to the case $n = \infty$.

We call the regime between the highly over-parameterized and under-parameterized regimes the *mildly over-parameterized* regime. The key question of interest is how the mildly over-parameterized regime bridges the highly over-parameterized regime and the under-parameterized regime. We have already seen that these two regimes differ in several aspects. One is that in the highly over-parameterized regime, the inner layer coefficients $\boldsymbol{b}$ barely change. In the under-parameterized regime, a small number of neurons experience large changes for their inner-layer coefficients.

First, we investigate the GD dynamics for two interesting scalings: $m = cn/(d+1)$ and $m = Cn$ where $c$ and $C$ are constants. Shown in Figures 6 are two examples for $m = 3n/(d+1)$ and $m = 0.75n$ with $n = 200, d = 19$ respectively. More examples with the same scalings but different values of $n$ can be found in Appendix D.

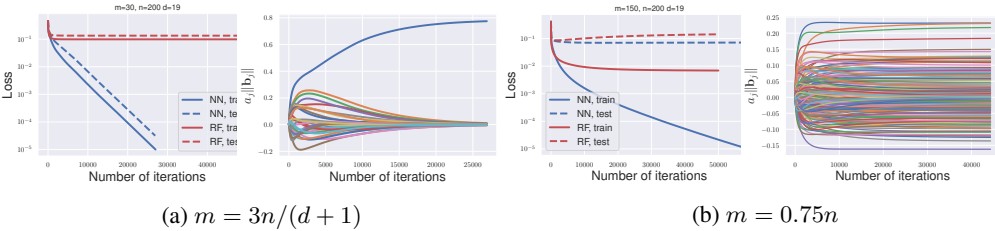

(a) $m = 3n/(d+1)$     (b) $m = 0.75n$

Figure 6: The dynamic behavior of learning single-neuron target function for two mildly over-parameterized cases. Here $n = 200, d = 19$ and learning rate is $0.001$. For each case, **Left:** The dynamics of the training and test losses (results from the corresponding random feature model is also plotted for comparison); **Right:** The dynamics of the magnitude of each neuron.

One can see that the behavior shown in Figures 6a resembles the ones shown for infinite samples, whereas the behavior shown in Figures 6b does not. In the first case, the test accuracy improves substantially after the GD dynamics departs from that of the RFM, and there is a notable presence of the activation phenomena. In the second case, the test error saturates soon after the GD dynamics

departs from that of the RFM, and there are no clear presence of the activation phenomena. We will call the first case "NN-like" and the second case "RF-like".

## 4.2 GENERALIZATION ERROR AND THE PATH NORM

Next we examine the generalization error of the GD solution. Shown in Figure 7 are the test errors of GD solutions with varying $m$'s and $n$'s. The target function is the single neuron target function $f_1^*$. We see that the test errors exhibit a sharp transition somewhere between $m = n/(d+1)$ to $m = n$. This is consistent with the observation in Figure 6 that when $m \sim n/(d+1)$, GD shows a NN-like behavior while $m \sim n$, GD shows a RF-like behavior. Moreover, the transition seems to become sharper with increasing values of $n$ and $m$.

In the following, we take a closer look at how the increase of $m$ affects the test performance by considering the circle neuron target function. The results for the single-neuron target function are similar, and can be found in Appendix E. For 2LNNs, the generalization gap can be controlled by the path norm of the parameters (Neyshabur et al., 2015; E et al., 2019b), defined by $\|\theta\|_P = \sum_{j=1}^{m} |a_j| \|\boldsymbol{b}_j\|_2$. Therefore we will compute the path norm of the parameters selected by GD.

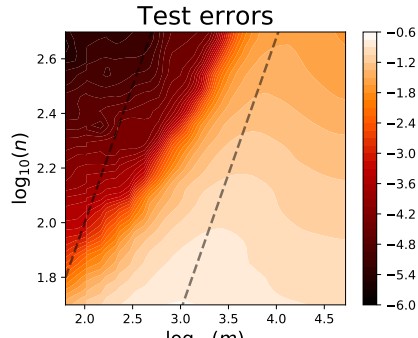

Figure 7: Heatmap of test errors of GD solutions with varying $m$'s and $n$'s. The target function is the single-neuron target function $f_1^*$ with $d = 40$. GD is stopped when the training loss is smaller than $10^{-8}$. The two dashed lines corresponds to $m = n/(d+1)$ and $m = n$, respectively.

Figure 8a examines the test error and the path norm as $m$ changes. One can see that as $m$ becomes larger, the test error of the NN model eventually becomes close to that of the RFM. One thing to notice is that these changes seem to behave smoothly across the points: $m = n/(d+1)$ and $m = n$, where the NN and the RFM change from an under-parameterized situation to an over-parameterized situation, respectively. We do not observe the peak of test errors around $m = n/(d+1)$ as suggested in (Belkin et al., 2019). We suspect that the peak observed in (Belkin et al., 2019) is the result of the special training method used there. Interestingly, Figure 8 and Figure 7 show that there does exist a peak around $m = n$, the same place for the RFM. This should be the result of the close proximity of the GD dynamics for 2LNN and RFM during the first phase. The latter performs extremely badly when $m = n$ due to resonance as shown in (Ma et al., 2020). Thus it is not surprising to see that NN also performs the worst around $m = n$.

In addition, the path norm seems to serve as a good indicator of the generalization performance. For example, one can see a dramatic increase of the path norm from $m = n/(d+1)$ to $m = n$, and the path norm peaks around $m = n$.

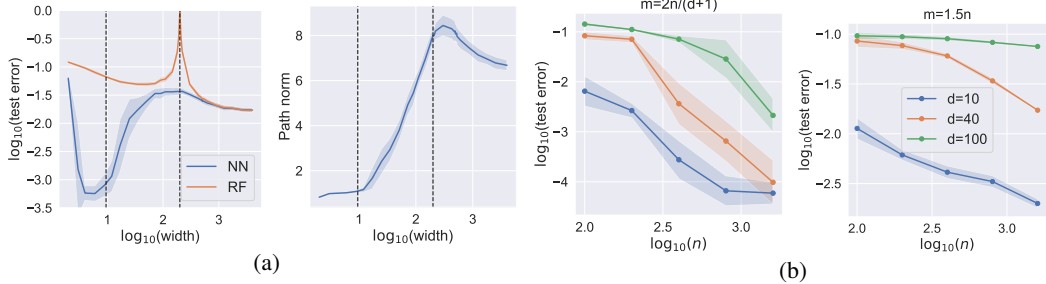

Figure 8: Test performance of GD solutions for the circle neuron $f_2^*$. (a) The path norm and test error as a function of $m$. Here $n = 200, d = 20$ and the learning rate is 0.001. GD is stopped after the training error is smaller than $10^{-6}$. (b) The test error as a function of $n$ for three different input dimensions.

To further explore the difference between the two scalings $m \sim n/(d+1)$ and $m \sim n$, we show in Figure 8b the test error as a function of the size of training set. The result suggests that under the scaling $m \sim n$, the test error may suffer from curse of dimensionality (CoD), while for the scaling

$m \sim n/(d + 1)$, it does not seem to be the case. Let the test error decrease as $O(1/n^{\alpha(d)})$. CoD means that $\alpha(d)$ decreases (to 0) as $d \to \infty$. Thus, it is difficult to learn the target function when $d$ is large if the model suffers from CoD.

## 5   GD DYNAMICS WITH MEAN-FIELD SCALING

For simplicity, we call the GD dynamics of the *scaled* 2LNN (2) GD-MF and the GD dynamics of the *unscaled* 2LNN (1) GD-conventional. Under the mean-field scaling, $a_j$ and $\boldsymbol{b}_j$ are of the same order. Thus $\dot{a}_j$ and $\dot{\boldsymbol{b}}_j$ are also of the same order, which means that there is no time-scale separation.

First let us look at the case when $n = \infty$. Figure 9 shows the dynamic behavior of GD-MF for single neuron target function $f_1^*$. Different from GD-conventional shown in Figure 1, we see that almost all the neurons move significantly and contribute to the model.

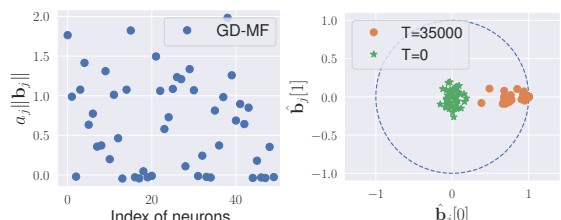

Figure 10 shows the test error for the case with finite sample. We see that the test error of GD-MF varies more smoothly with the increase of network width. There is almost no clear deterioration of the performance even when we increase the network width to the highly over-parameterized regime. This is clearly different from the case for GD-conventional. Moreover, we observe that the GD-MF solutions do not

Figure 9: GD-MF dynamics for learning the single neuron target function $f_1^*$. Here $m = 50, d = 100, n = \infty$ and learning rate is 0.001. **Left:** The magnitude of each neuron for the converged solution. **Right:** The projection to the first two coordinates of $\hat{\boldsymbol{b}}$ for each neuron. The green ones correspond to the random initialization; the orange ones correspond to the solutions found by GD-MF.

seem to suffer from CoD at the regime $m \sim n$, which is not the case for GD-conventional (see Figure 8b).

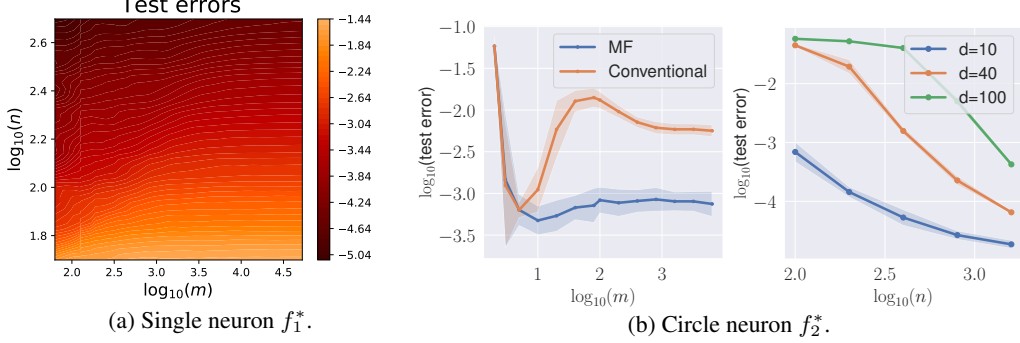

(a) Single neuron $f_1^*$.                           (b) Circle neuron $f_2^*$.

Figure 10: (a) The heatmap of test errors of GD-MF solutions for learning the single neuron $f_1^*$. (b) The test performance of learning the circle neuron $f_2^*$. The left shows the test error as a function of $m$. The right shows the test error as a function of $n$ for the scaling $m = 1.5n$.

## 6   CONCLUSION

The experimental results shown in this paper suggest the following: (1) Under the conventional scaling, both the training process and the generalization performance are quite sensitive to the network parameters. With the mean-field scaling, both are more stable, though not always better. (2) In the NN-like regime under conventional scaling, the quenching process provides the mechanism of implicit regularization, since it does not allow the the path norm to grow out of control. Consequently the generalization gap is controlled.

Obviously, many questions remain open. For example, how do the results discussed in this paper manifest for practical datasets and deep neural networks? Can we substantiate some of the findings by rigorous results? These are all questions that are left for future work.

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

# A    THE INFLUENCE OF THE INITIAL MAGNITUDE OF THE OUTER COEFFICIENTS

First we consider the case $\beta = 1/\sqrt{m}$. Figure 11 shows the numerical results with the same setting as in Section 3.1. We see that the dynamical behavior is qualitatively the same as the case of $\beta = 0$.

For $\beta = 1/m^\gamma$ with $\gamma \geq 1/2$, the time-scale separation between the inner and outer layers always hold initially as long as $m$ is large enough. In these cases we see basically the same kind of behavior as shown in the main text. To simplify the experiments and the presentation, we set $\beta = 0$, i.e. $\gamma = \infty$, in which case we do not need to take very large values of $m$.

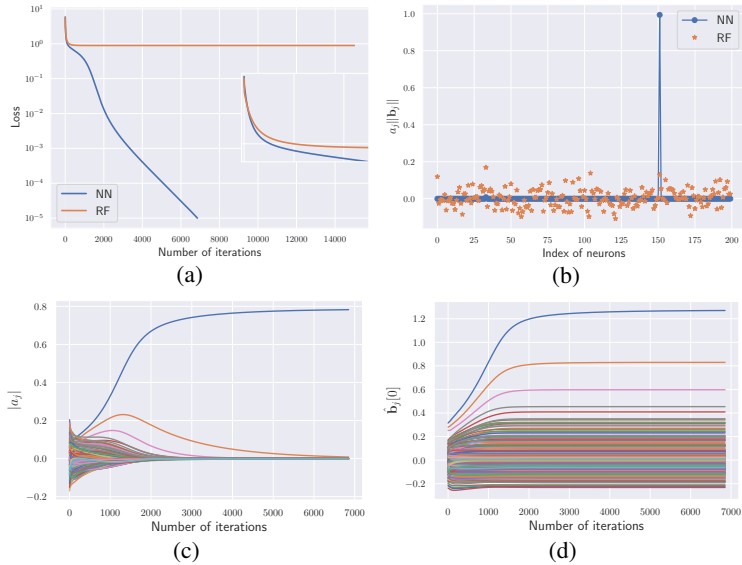

Figure 11: The dynamic behavior of GD with $\beta = 1/\sqrt{m}$. Here the target function is the single neuron $f_1^*$. $m = 200, d = 100$ and learning rate is $0.001$. We also plot the results of RFM as comparison. (a) The dynamic behavior of the population risk; (b) The magnitude of each neuron for the converged solution. (c) The dynamics of the $a$ coefficient of each neuron. (d) The dynamics of $\{\hat{\boldsymbol{b}}_j[0]\}$, the projection of $\hat{\boldsymbol{b}}_j$ to $\boldsymbol{b}^* = \boldsymbol{e}_1$.

# B    ADDITIONAL RESULTS FOR THE FINITE NEURON TARGET FUNCTION

Figure 12 shows additional results for learning finite neurons when $m$ is much larger than $m^*$. We see that the behavior is basically the same as Figure 3 except that more neurons concentrate around $0$.

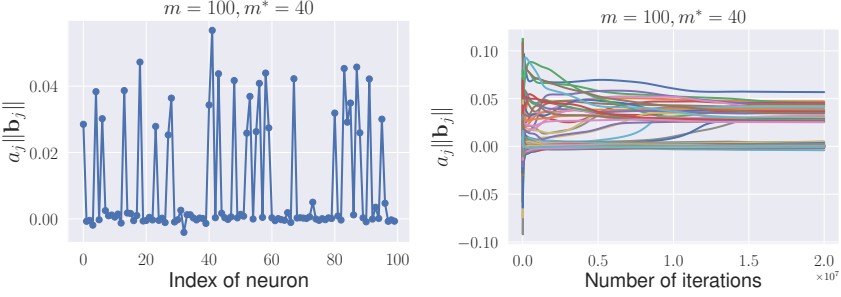

Figure 12: Learning finite neurons with $a_j^* = 1/m, \boldsymbol{b}_j^* \sim \pi_0$. Here $m = 100, m^* = 40$ and learning rate is $0.001$. **Left:** The magnitude of each neuron of the final solution; **Right:** The dynamics of the magnitude for each neuron.

## C   ADDITIONAL RESULTS FOR THE EFFECTIVE DYNAMICS

In Figure 13, we plot the difference between the effective dynamics and original dynamics as a function of number of iterations. We see that the two dynamics are pretty close in the second phase even in a long time scale.

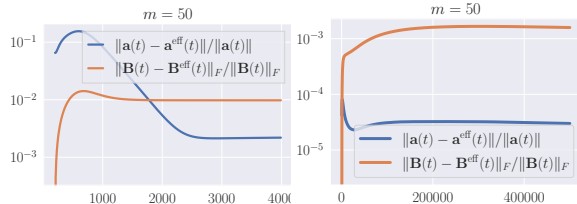

Figure 13: The difference between the effective dynamics and the original dynamics. **Left:** The single neuron target function $f_1^*$ with $m = 50$; **Right:** The surface neuron target function $f_4^*$ with $m = 50$.

.

## D   ADDITIONAL EXPERIMENTAL RESULTS FOR GD DYNAMICS WITH FINITE SAMPLES

### D.1   THE UNDER-PARAMETERIZED REGIME

We first look at the regime where the network is under-parameterized, i.e. $m < n/(d+1)$. A typical result for the single neuron target function is shown in Figure 14. We see that overall the qualitative behavior of GD is similar to that of the case when $n = \infty$ studied in Section 3. There are still two phases, and the convergent solution is sparse. Figure 14b shows that the quenching process becomes slower compared to the case with infinite data.

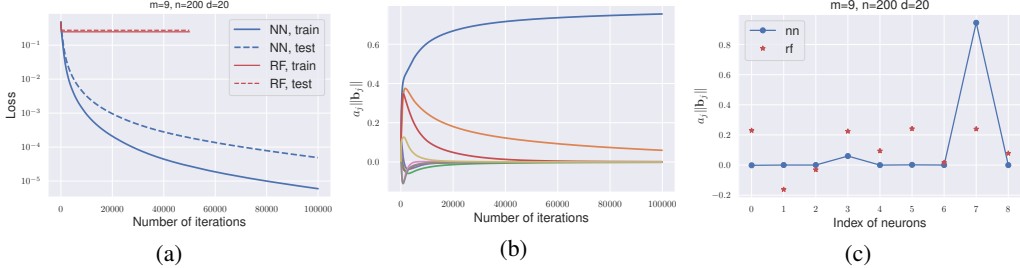

Figure 14: GD dynamics for learning the single neuron target function in the under-parameterized regime. Here $m = 9, d = 20, n = 200$ and the learning rate $\eta = 0.005$. (a) Time history of training and test errors. (b) Time history of the magnitude for each neuron. (c) The magnitude of each neuron of the convergent solution.

Figure 15 shows the result for the circle neuron target function. We see again that there are still two phases. But the multi-step phenomenon becomes less pronounced compared to the infinite data case. From Figure 15b, the neurons can still roughly be divided into two groups: active and quenched neurons. Also we still observe some activation during the second phase, although this process is much more smooth compared to the case with infinite data.

### D.2   THE MILDLY OVER-PARAMETERIZED REGIME

Here we provide more experimental results under the two scalings $m \sim n/(d+1)$ and $m \sim n$. The results are shown in Figure 16 and 17. We see that the dynamic behavior is similar to the ones in Figure 6.

### D.3   THE HIGHLY OVER-PARAMETERIZED REGIME

We first look at the simple situation where the network is highly over-parameterized. Figure 18 shows some typical results in this regime. Clearly in this regime, the GD dynamics for the NN model stay uniformly close to that of the RFM for all time, as was proved in E et al. (2020). We

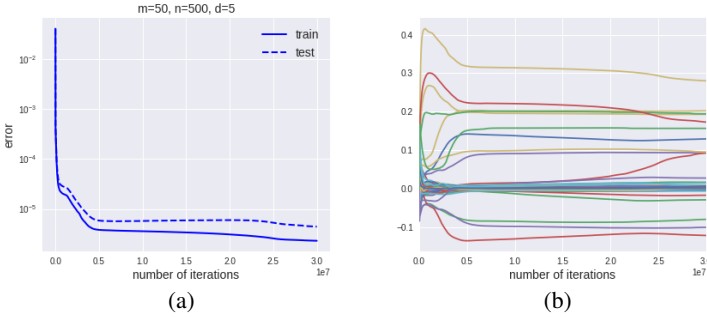

Figure 15: GD dynamics for the circle neuron target function in the under-parameterized case. Here $m = 50, n = 500, d = 5$. (a) The dynamics of training and test loss. (b) The dynamics of the outer coefficients.

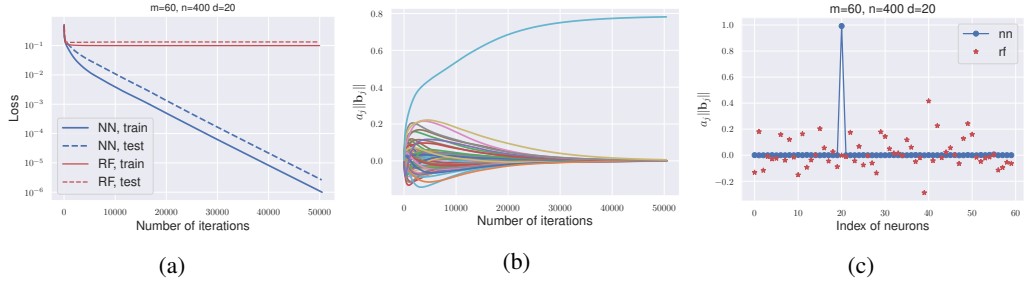

Figure 16: The dynamic behavior of the GD solutions for $m = 3n/(d + 1)$. Here $m = 60, n = 400, d = 19$ and learning rate $\eta = 0.001$.

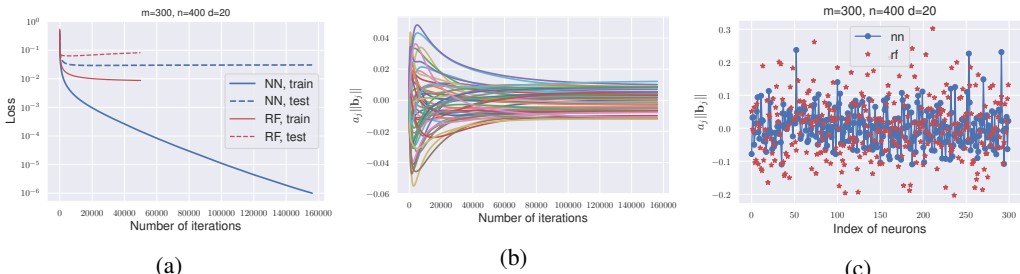

Figure 17: The dynamic behavior of the GD solutions for $m = 0.75n$. Here $m = 300, n = 400, d = 19$ and learning rate $\eta = 0.001$.

see that the training error goes to 0 exponentially fast, but the testing error quickly saturates. Note that Theorem 1 suggests that the network width should satisfy $m \gtrsim n^2 \lambda_n^{-4} \ln(n^2 \delta^{-1})$, but in this experiment $m = 0.5n^2$ is enough.

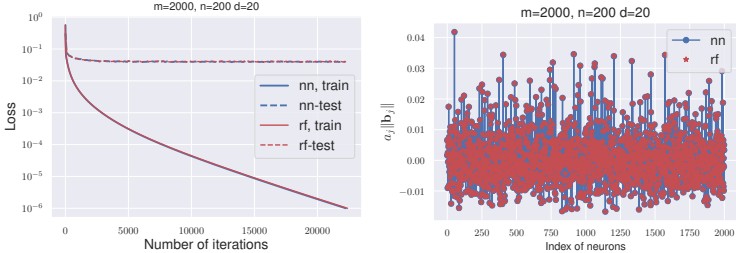

Figure 18: Comparison between the GD solutions for 2LNN and RFM for the single neuron target function $f_1^*$. The learning rate 0.001 and $m = 2000, n = 200, d = 20$. **Left:** The time history of the training and test error. **Right:** The magnitude of the converged solutions.

# E  ADDITIONAL RESULTS ON THE TEST PERFORMANCE

Figure 19 shows how the test error is affected by the number of samples $n$ and the number of neurons $m$ for the single neuron target function.

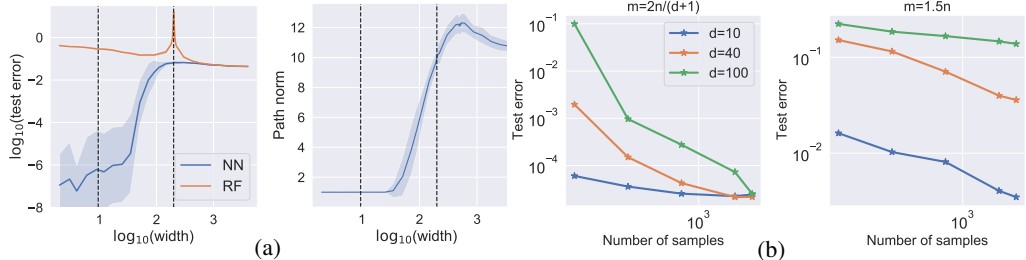

Figure 19: Test performance of GD solutions for single neuron $f_1^*$. (a) The path norm and test error as a function of $m$. Here $n = 200, d = 20$ and the learning rate is 0.001. GD is stopped after the training error is smaller than $10^{-8}$. (b) The test errors as a function of $n$ for three different input dimensions.

