# OpenReview forum: "The Quenching-Activation Behavior of the Gradient Descent Dynamics for Two-layer Neural Network Models"
_ICLR.cc/2021/Conference — Reject_

### Official Review · AnonReviewer1 · 2020-10-27
**Interesting empirical observation; lack comments on the neural tangent scaling; lack mathematical support.**

**Rating:** 5
**Confidence:** 4

**Review:**

This paper studies the gradient dynamics of two-layers neural networks. It is empirically shown that, for a specific type of initialization, for less over-parameterized neural networks, the gradient dynamics follows two phases: a phase that follows the random features model where all the neurons are "quenched", and another phase in which there are a few "activated" neurons. This type of behavior of gradient dynamics is different from that of the mean-field regime.

This paper is well-written. The quenching and active phenomena is interesting. Existing theory characterized the behavior of neural networks in the mean-field/neural tangent regime, and this paper studied a regime that is different from the existing ones. The authors used the effective dynamics to explain the second phase of the dynamics, which gives some good intuitions.

There are the following issues of this paper.
1. This paper did not distinguish between "random features model" and "neural tangent model", although they are closely related. In particular, Theorem 1 presented in this paper is connected to the "random features model" as in [E, et.al. 2020], which does not really give the "neural tangent kernel" in [Jacot, et.al. 2018] and [Du, et.al., 2019b]. The difference is that, "random features model" only the linearization of the top layer, while the "neural tangent kernel" also contains the linearization of the bottom layer. Although they are closely connected, it is important to clearly point out the distinction. Moreover, the "neural tangent" scaling is a different scaling (different than both MF and RF scaling), where there is a 1/\sqrt{m} coefficient in front of the network. I think at least the authors should discuss what happens for the "neural tangent" scaling when it is less over-parameterized.
2. This is a pure empirical paper and the phenomenon has no justification (even heuristic mathematical justification). It would be good if the authors can (heuristically) explain which neuron will become activated (any random neuron, or a neuron that satisfies some initial condition?), and how long it will take for those neurons to become activated (perhaps this depends on the initial weights of the neurons). Without these justifications, I feel that the paper is less complete. (In comparison, the MF and NT results are rigorously established. )

Overall, I think the phenomenon discovered in this paper is interesting. However, it didn't comment upon the neural tangent scaling, and it lacks mathematical support. I feel that this paper is on the borderline. I will consider raising my score if the authors can resolve these issues.

---

### Official Review · AnonReviewer3 · 2020-10-27
**Interesting finding, but not a complete picture**

**Rating:** 5
**Confidence:** 4

**Review:**

The paper experimentally identifies a phenomenon in which the participation among the neurons in a two-layer neural net trained with gradient descent (GD) changes with time. In particular:
- The paper shows for certain artificial target functions with low-dimensional structures, the neurons start out from a random-feature behavior and then a few are selectively activated, distinct from the rest of the neurons which are “suppressed”.
- This behavior is shown to be pronounced when the network is less (or mildly) over-parameterized and suffers less from the curse of dimensionality (CoD). It is almost unobservable when the network has width m exceeding the training size n and suffers more from CoD.
- Lastly the paper argues that this behavior is specific to the choice of scaling, by showing that under the mean field scaling, the network displays much less sensitivity to the scale of the width.

Large-width neural networks have been a central object of studies for the past few years. With certain scalings (typically collectively referred to as NTK scalings), as width goes to infinity, the network becomes “linearized”. This is probably among the few works that studies how the network under NTK scaling can behave beyond the linearized regime. In particular, while most of these few works concern with the mathematical question of “how much” the network can learn better than linearized methods, this work identifies a novel mechanism on “how different” the network’s behavior is from linearized methods. I’m unaware if the quenching/activation mechanism has been reported before. In this sense, the result of the paper has an interesting position in this literature.

The measurement of a neuron’s participation via the product of the two weights is interesting and elucidative of the use of the path norm. The presentation is also clear and easy to follow.

The paper however has several shortcomings. In no particular order:

1) The target functions are idealized, and no experiments are reported on more realistic settings. Is the quenching/activation mechanism a common phenomenon with real life datasets? Are the transition boundaries n/(d+1) and n universal? We observe from Fig. 4 that no clear quenching/activation process takes place, so it’s questionable whether it can be observed for a difficult and highly diverse real life dataset. A suggestion is to perform experiments on MNIST, CIFAR-10 and CIFAR-100 (which might resemble the single neuron, circle neuron and surface neuron examples).

2) The paper may consider having a discussion on recent works (e.g. [1, 2]) that consider linearized behaviors in the regime where m depends mildly on n/d or is almost independent of n, and whether they may conflict with the result in the paper.

3) The paper gives no explanation or heuristic to understand how the quenching/activation phenomenon arises. In particular, can we know in advance which neuron is going to be activated, even in the simple case of single neuron target function? The paper mentions “The same behavior was observed with other realizations of the initial data, except that the number of activated neurons can be different.” Does it mean that it is entirely possible, although unlikely, that all neurons are activated? If the size of the activated neuron set is highly unpredictable, it would limit usefulness of the effective dynamics.

4) It is unclear how to make the effective dynamics a useful tool. It looks like the switching time (at which we switch from the first phase to the second phase) and the sets I1 and I2 have to be observed a priori from the experiment with the full dynamics. Is there an understanding on simple properties (e.g. how large) of these quantities?

5) In the derivation of the effective dynamics, it is unclear why the second layer’s weights of the quenched neurons move faster than the first layer’s weights. We only know from Theorem 1 that such timescale separation occurs when the network is random-feature-like, but we do not know if it is so during the quenching process in the mildly over-parameterized and under-parameterized regimes.  One can actually observe from Fig. 1 that most neurons are still not quenched when the network departs from the random-feature behavior.

6) The paper suggests that due to quenching of most neurons, the path norm remains small since it is essentially a sum over only activated neurons. However the circle neuron target example shows that the population risk continues to decrease once a neuron emerges from the quenched state and becomes activated. Does this somehow violate the path norm intuition? How does the path norm look like through time?

Ultimately I find the finding interesting, but the paper leaves a lot of questions unanswered. Still I would encourage the authors substantiate the paper with more analyses and findings, empirical or theoretical, since this is quite a novel finding.

[1] Polylogarithmic width suffices for gradient descent to achieve arbitrary small test error with shallow relu networks, Ji and Telgarsky, ICLR 2020.

[2] Neural networks Learning and Memorization with (almost) no Over-Parameterization, Daniely, 2019.

---

### Official Review · AnonReviewer4 · 2020-10-28
**Large set of experiments revealing new observations on gradient descent dynamics in a simple, controlled setting**

**Rating:** 5
**Confidence:** 4

**Review:**

This paper studies gradient descent dynamics of two-layer neural networks with ReLU activation function. In a controlled setting, a list of experiments investigates the dynamics and compares them for different regimes where the relation of number of hidden neurons to number of training samples is changed and for several specific target functions. The paper proposes that training happens in two steps: The first phase resembles the learning of a random feature model (RFM) where only the output layer coefficients are changed. The second phase can activate or deactivate neurons and decreases coefficients to favor sparse solutions if the target function admits a sparse solution. The dynamics are further compared the setting of mean-field scaling, which shows a different learning dynamic.

Pros:
+ The paper presents experiments on dynamics of gradient descent with the goal to give new insight on the important and unsolved phenomenon of implicit bias observed in deep neural networks.
+ By starting in a (simplified) controlled setting, a large scope of experiments is conducted that studies different aspects and different training regimes. The setting and the experiments are all well-described and clearly presented. Interesting behavior of the learning dynamics can therefore be shown.
+ With the largely overparameterized setting fairly well-understood, the current paper approaches the medium-overparameterized regime where the learning dynamics are more complicated and more close to neural networks in practice. Therefore, the paper targets an important knowledge gap.

While some interesting observations can be made, I consider the presented results and gained knowledge as limited in scope and hence slightly tend to suggest to reject.

Cons:
- In particular, no clear conclusion can be made from the results. In fact, there is no conjecture of how the results may manifest in practical neural networks. (In case I missed such a conjecture, then I wonder why the conjecture was not tested in a practical setting.)
- The paper is purely experimental. Since it considers largely simplified settings (specific target functions, data sampled uniformly from the unit sphere, in most experiments the true solution can be found with a sub-network consisting of only one hidden neuron) and since the networks and optimization method (gradient descent) are rather simple, one could expect that at least some theoretical contribution or explanation could be given, which the paper lacks entirely.
- While there is a large scope of experiments conducted, some results are only partial and draw conclusions without further investigations. It is hard to tell from the presented results how characteristic they are for other settings or behave under small changes such as the input dimension, target function, loss function, etc.
Further more detailed remarks on the specific contributions and the presentation are below.

For the author’s reply, I would appreciate a clarification of the exact contributions summarizing the findings and its possible implications for the training of practical neural networks.


Comments on the specific contributions:
- The first contribution points out the existence two phases. In the first phase, the weights of the first layer do not change and the model behaves like a random feature model. This phenomenon seems to be limited to a few iterations where it is an almost trivial observation considering that the second-layer weights are initialized at zero (which implies a vanishing gradient for first layer weights).
- The third mentioned contribution is peculiar since the only change is a factor of 1/m to the network function, which is equivalent (for the considered squared loss) to a scaling of the target function. The stated contribution is therefore that this scaling changes the observed dynamics significantly, which casts doubt when the observations generalize to other settings.
- The observed behavior aims to explain the implicit bias of neural networks. However, this behavior can also be observed in overparamterized networks and the observed dynamics differ in the considered regimes of under-and overparameterized networks. This poses the question in which way the implicit bias is explained by the observations.
 - The paper suffers from its presentation. The list of contributions mentions an observed transition (what kind of transition?) and consider undefined terminology of quenched neurons. The terminology of quenched neurons is only loosely explained much later on page 4 as neurons that are “quenched” in the sense that their outer layer coefficients eventually start decreasing and then keep decreasing. (Shall quenching neurons have coefficients that converge to zero? If neurons can still pop up at a later stage, are they still considered quenched neurons? Are neurons with constant output layer coefficients but norm-decreasing inner layer coefficients also quenching (this difference cannot be distinguished in the plots)? It would be good to make precise what the authors mean by quenched neurons.


More detailed remarks:
- First line page 3: citation seems misplaced, as I was unable to find the result in the paper. Please update reference or show a proof.
- How many iterations does the first phase consist of where the model neural network behaves like a RFM?
- Is it correct that the two phases in the loss development can not consistently be matched with changes in the parameters?
- Is it correct that in the most realistic setting 3.1.2, there is barely a quenching behavior visible?
- I was wondering about the notion of overparameterization. Why would a network be underparameterized just because the width is small even if the target function can be learned with a single neuron?It sounds reasonable to consider the setting of training polynomials of maximal degree 30 to learn a linear function as an overparameterized problem, no matter how many samples are considered for training.
- The consideration to compare m being proportional to n against m being proportional to n/(d+1) has little meaning without fixing or comparing the respective constants (m=10^9*n/(d+1) vs m=10^-9*n also satisfies the stated proportionality), and most importantly without experimenting with different dimensions d. Since it is not investigated how the behavior changes with changes of d, there cannot be made any conclusion about the different regimes. Also the consistency claim in 4.2 is meaningless when changing the constants. It would be consistent if in 4.1 one would consider m=n and m=n/(d+1) instead of these terms scaled by constants (For a single d, suitable constants can always be found.) If no experiments for changing input dimension are carried out, then all we need to care about is the quotient m/n.


Typos:
Figure 6: legend not entirely visible
Conclusion: „the the“

---

### Official Review · AnonReviewer5 · 2020-11-05

**Rating:** 5
**Confidence:** 4

**Review:**

**SUMMARY** The authors perform an empirical study of the dynamics of gradient decent for single-hidden-layer NNs with ReLU activations on some synthetic datasets under L2 loss. They identify two distinct phases: an early phase matching RF models and a late phase where the neurons split into activated and quenched groups. When the target function admits a sparse representation, the learned function can have a sparse activation pattern, which is interpreted as a form of implicit regularization.

**PROS** The authors identify a mechanism of quenching and sparse activation patterns that should be of interest to researchers studying the generalization properties of NNs. The mechanism seems sensitive to the data distribution and the network width

**CONS** My main concern is about the robustness of the results. Several simplifying or arbitrary choices are made, which while necessary for theoretical analysis, should be analyzed more critically in an empirical paper. Specifically, the results are restricted to a simple single-hidden-layer architecture, ReLU activation functions, and a simplistic, synthetic data distribution.

 **RECOMMENDATION** In its current form, I vote for rejecting the paper. While there are some interesting observations, the uncertainty about their robustness and the absence of a theoretical explanation weaken the paper. Addressing either of these shortcomings would improve the paper.

  **ADDITIONAL QUESTIONS**
- The scaling for the first-layer weights b seems odd and nonstandard. Since the data are points on the unit sphere, the scaling I/d means the pre/post-activations are little-o 1. Is this intentional? What happens if the b_ij are order 1?
- How are \mathbb{I}_1 and \mathbb{I}_2 determined in Sec. 3.2 and Fig. 5?
- How correlated are the test error and the path norm? In Fig. 8 they look related, but it’s difficult to tell how much.
- Is the step behavior in Fig. 2 related to Ghorbani, Mei et al. 2020?

**MINOR COMMENTS**
- Some additional motivation for f* when pi* is uniform on the unit circle would be helpful.
- Consider including some additional citations for the random feature model, particularly in Sec. 4.1: Mei and Montanari 2019; d'Ascoli, Refinetti et al. 2020; Adlam and Pennington 2020. They study the effects of under/overparameterization on the generalization error.
- I also think the overall structure of the paper could be improved to support the main claim the paper is making.

---

### Decision · Program_Chairs · 2021-01-07
**Final Decision**

**Decision:**

Reject

**Comment:**

This paper empirically investigates the gradient dynamic of two-layer network nets with ReLU activations on synthetic datasets under $L^2$ loss. The empirical results show that for a specific type of initialization and less overparametrized neural nets, the gradient dynamics experience two phases: a phase that follows the random features model where all the neurons are *quenched* and another phase where there are a few *activated* neurons. As pointed out by Reviewer 1, this paper lacks mathematical support and did not distinguish between *random features model* and *neural tangent model*. Reviewer 3 and Reviewer 4 also complained that the paper is purely experimental. Therefore, this paper may benefit from proposing an at least heuristic or high-level conjecture/interpretation/argument that tries to explain the empirical results.